# Scaling Gaussian Process Regression with Derivatives

**David Eriksson**
Center for Applied Mathematics
Cornell University
Ithaca, NY 14853
dme65@cornell.edu

**Kun Dong**
Center for Applied Mathematics
Cornell University
Ithaca, NY 14853
kd383@cornell.edu

**Eric Hans Lee**
Department of Computer Science
Cornell University
Ithaca, NY 14853
ehl59@cornell.edu

**David Bindel**
Department of Computer Science
Cornell University
Ithaca, NY 14853
bindel@cornell.edu

**Andrew Gordon Wilson**
School of Operations Research
and Information Engineering
Cornell University
Ithaca, NY 14853
andrew@cornell.edu

## Abstract

Gaussian processes (GPs) with derivatives are useful in many applications, including Bayesian optimization, implicit surface reconstruction, and terrain reconstruction. Fitting a GP to function values and derivatives at $n$ points in $d$ dimensions requires linear solves and log determinants with an $n(d+1) \times n(d+1)$ positive definite matrix – leading to prohibitive $\mathcal{O}(n^3 d^3)$ computations for standard direct methods. We propose iterative solvers using fast $\mathcal{O}(nd)$ matrix-vector multiplications (MVMs), together with pivoted Cholesky preconditioning that cuts the iterations to convergence by several orders of magnitude, allowing for fast kernel learning and prediction. Our approaches, together with dimensionality reduction, enables Bayesian optimization with derivatives to scale to high-dimensional problems and large evaluation budgets.

## 1 Introduction

Gaussian processes (GPs) provide a powerful probabilistic learning framework, including a *marginal likelihood* which represents the probability of data given only kernel hyperparameters. The marginal likelihood automatically balances model fit and complexity terms to favor the simplest models that explain the data [22, 21, 27]. Computing the model fit term, as well as the predictive moments of the GP, requires solving linear systems with the kernel matrix, while the complexity term, or *Occam's factor* [18], is the log determinant of the kernel matrix. For $n$ training points, exact kernel learning costs of $\mathcal{O}(n^3)$ flops and the prediction cost of $\mathcal{O}(n)$ flops per test point are computationally infeasible for datasets with more than a few thousand points. The situation becomes more challenging if we consider GPs with both function value and derivative information, in which case training and prediction become $\mathcal{O}(n^3 d^3)$ and $\mathcal{O}(nd)$ respectively [21, §9.4], for $d$ input dimensions.

Derivative information is important in many applications, including Bayesian Optimization (BO) [29], implicit surface reconstruction [17], and terrain reconstruction. For many simulation models, derivatives may be computed at little extra cost via finite differences, complex step approximation, an adjoint method, or algorithmic differentiation [7]. But while many scalable approximation methods for Gaussian process regression have been proposed, scalable methods incorporating derivatives have received little attention. In this paper, we propose scalable methods for GPs with derivative information built on the *structured kernel interpolation* (SKI) framework [28], which uses local interpolation to map scattered data onto a large grid of inducing points, enabling fast MVMs using FFTs. As the uniform grids in SKI scale poorly to high-dimensional spaces, we also extend the structured kernel interpolation for products (SKIP) method, which approximates a high-dimensional product kernel as a Hadamard product of low rank Lanczos decompositions [8]. Both SKI and SKIP provide fast approximate kernel MVMs, which are a building block to solve linear systems with the kernel matrix and to approximate log determinants [6].

The specific contributions of this paper are:

- We extend SKI to incorporate derivative information, enabling $\mathcal{O}(nd)$ complexity learning and $\mathcal{O}(1)$ prediction per test points, relying only on fast MVM with the kernel matrix.

- We also extend SKIP, which enables scalable Gaussian process regression with derivatives in high-dimensional spaces without grids. Our approach allows for $\mathcal{O}(nd)$ MVMs.

- We illustrate that preconditioning is critical for fast convergence of iterations for kernel matrices with derivatives. A pivoted Cholesky preconditioner cuts the iterations to convergence by several orders of magnitude when applied to both SKI and SKIP with derivatives.

- We illustrate the scalability of our approach on several examples including implicit surface fitting of the Stanford bunny, rough terrain reconstruction, and Bayesian optimization.

- We show how our methods, together with active subspace techniques, can be used to extend Bayesian optimization to high-dimensional problems with large evaluation budgets.

- Code, experiments, and figures may be reproduced at:
  `https://github.com/ericlee0803/GP_Derivatives`.

We start in §2 by introducing GPs with derivatives and kernel approximations. In §3, we extend SKI and SKIP to handle derivative information. In §4, we show representative experiments; and we conclude in §5. The supplementary materials provide several additional experiments and details.

## 2  Background and Challenges

A Gaussian process (GP) is a collection of random variables, any finite number of which are jointly Gaussian [21]; it also defines a distribution over functions on $\mathbb{R}^d$, $f \sim \mathcal{GP}(\mu, k)$, where $\mu : \mathbb{R}^d \to \mathbb{R}$ is a mean field and $k : \mathbb{R}^d \times \mathbb{R}^d \to \mathbb{R}$ is a symmetric and positive (semi)-definite covariance kernel. For any set of locations $X = \{x_1, \ldots, x_n\} \subset \mathbb{R}^d$, $f_X \sim \mathcal{N}(\mu_X, K_{XX})$ where $f_X$ and $\mu_X$ represent the vectors of function values for $f$ and $\mu$ evaluated at each of the $x_i \in X$, and $(K_{XX})_{ij} = k(x_i, x_j)$. We assume the observed function value vector $y_X \in \mathbb{R}^n$ is contaminated by independent Gaussian noise with variance $\sigma^2$. We denote any kernel hyperparameters by the vector $\theta$. To be concise, we suppress the dependence of $k$ and associated matrices on $\theta$ in our notation. Under a Gaussian process prior depending on the covariance hyperparameters $\theta$, the log marginal likelihood is given by

$$\mathcal{L}(y_X \,|\, \theta) = -\frac{1}{2}\left[(y_X - \mu_X)^T \alpha + \log|\tilde{K}_{XX}| + n \log 2\pi\right] \tag{1}$$

where $\alpha = \tilde{K}_{XX}^{-1}(y_X - \mu_X)$ and $\tilde{K}_{XX} = K_{XX} + \sigma^2 I$. The standard direct method to evaluate (1) and its derivatives with respect to the hyperparameters uses the Cholesky factorization of $\tilde{K}_{XX}$, leading to $\mathcal{O}(n^3)$ kernel learning that does not scale beyond a few thousand points.

A popular approach to scalable GPs is to approximate the exact kernel with a structured kernel that enables fast MVMs [20]. Several methods approximate the kernel via *inducing points* $U = \{u_j\}_{j=1}^m \subset \mathbb{R}^d$; see, e.g. [20, 16, 13]. Common examples are the subset of regressors (SoR), which exploits low-rank structure, and fully independent training conditional (FITC), which introduces an additional diagonal correction [23]. For most inducing point methods, the cost of kernel learning with $n$ data points and $m$ inducing points scales as $O(m^2 n + m^3)$, which becomes expensive as $m$

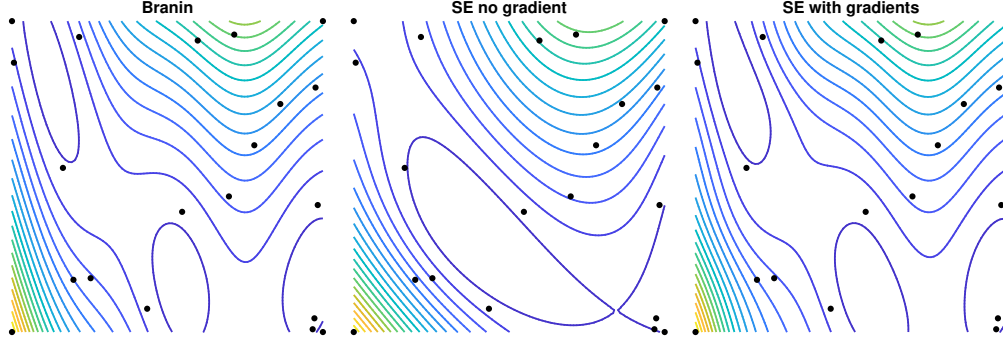

Figure 1: An example where gradient information pays off; the true function is on the left. Compare the regular GP without derivatives (middle) to the GP with derivatives (right). Unlike the former, the latter is able to accurately capture critical points of the function.

grows. As an alternative, Wilson and Nickisch [28] proposed the structured kernel interpolation (SKI) approximation,

$$K_{XX} \approx W K_{UU} W^T \qquad (2)$$

where $U$ is a uniform grid of inducing points and $W$ is an $n$-by-$m$ matrix of interpolation weights; the authors of [28] use local cubic interpolation so that $W$ is sparse. If the original kernel is stationary, each MVM with the SKI kernel may be computed in $\mathcal{O}(n + m \log(m))$ time via FFTs, leading to substantial performance over FITC and SoR. A limitation of SKI when used in combination with Kronecker inference is that the number of grid points increases exponentially with the dimension. This exponential scaling has been addressed by structured kernel interpolation for products (SKIP) [8], which decomposes the kernel matrix for a product kernel in $d$-dimensions as a Hadamard (elementwise) product of one-dimensional kernel matrices.

We use fast MVMs to solve linear systems involving $\tilde{K}_{XX}$ by the method of conjugate gradients. To estimate $\log|\tilde{K}_{XX}| = \mathrm{tr}(\log(\tilde{K}_{XX}))$, we apply stochastic trace estimators that require only products of $\log(\tilde{K}_{XX})$ with random probe vectors. Given a probe vector $z$, several ideas have been explored to compute $\log(\tilde{K}_{XX})z$ via MVMs with $\tilde{K}_{XX}$, such as using a polynomial approximation of $\log$ or using the connection between the Gaussian quadrature rule and the Lanczos method [11, 25]. It was shown in [6] that using Lanczos is superior to the polynomial approximations and that only a few probe vectors are necessary even for large kernel matrices.

Differentiation is a linear operator, and (assuming a twice-differentiable kernel) we may define a multi-output GP for the function and (scaled) gradient values with mean and kernel functions

$$\mu^\nabla(x) = \begin{bmatrix} \mu(x) \\ \partial_x \mu(x) \end{bmatrix}, \qquad k^\nabla(x, x') = \begin{bmatrix} k(x, x') & (\partial_{x'} k(x, x'))^T \\ \partial_x k(x, x') & \partial^2 k(x, x') \end{bmatrix},$$

where $\partial_x k(x, x')$ and $\partial^2 k(x, x')$ represent the column vector of (scaled) partial derivatives in $x$ and the matrix of (scaled) second partials in $x$ and $x'$, respectively. Scaling derivatives by a natural length scale gives the multi-output GP consistent units, and lets us understand approximation error without weighted norms. As in the scalar GP case, we model measurements of the function as contaminated by independent Gaussian noise.

Because the kernel matrix for the GP on function values alone is a submatrix of the kernel matrix for function values and derivatives together, the predictive variance in the presence of derivative information will be strictly less than the predictive variance without derivatives. Hence, convergence of regression with derivatives is always superior to convergence of regression without, which is well-studied in, e.g. [21, Chapter 7]. Figure 1 illustrates the value of derivative information; fitting with derivatives is evidently much more accurate than fitting function values alone. In higher-dimensional problems, derivative information is even more valuable, but it comes at a cost: the kernel matrix $K_{XX}^\nabla$ is of size $n(d+1)$-by-$n(d+1)$. Scalable approximate solvers are therefore vital in order to use GPs for large datasets with derivative data, particularly in high-dimensional spaces.

## 3 Methods

One standard approach to scaling GPs substitutes the exact kernel with an approximate kernel. When the GP fits values and gradients, one may attempt to separately approximate the kernel and the kernel derivatives. Unfortunately, this may lead to indefiniteness, as the resulting approximation is no longer a valid kernel. Instead, we differentiate the approximate kernel, which preserves positive definiteness. We do this for the SKI and SKIP kernels below, but our general approach applies to any differentiable approximate MVM.

### 3.1 D-SKI

D-SKI (SKI with derivatives) is the standard kernel matrix for GPs with derivatives, but applied to the SKI kernel. Equivalently, we differentiate the interpolation scheme:

$$k(x, x') \approx \sum_i w_i(x) k(x_i, x') \rightarrow \nabla k(x, x') \approx \sum_i \nabla w_i(x) k(x_i, x').$$

One can use cubic convolutional interpolation [14], but higher order methods lead to greater accuracy, and we therefore use quintic interpolation [19]. The resulting D-SKI kernel matrix has the form

$$\begin{bmatrix} K & (\partial K)^T \\ \partial K & \partial^2 K \end{bmatrix} \approx \begin{bmatrix} W \\ \partial W \end{bmatrix} K_{UU} \begin{bmatrix} W \\ \partial W \end{bmatrix}^T = \begin{bmatrix} W K_{UU} W^T & W K_{UU} (\partial W)^T \\ (\partial W) K_{UU} W^T & (\partial W) K_{UU} (\partial W)^T \end{bmatrix},$$

where the elements of sparse matrices $W$ and $\partial W$ are determined by $w_i(x)$ and $\nabla w_i(x)$ — assuming quintic interpolation, $W$ and $\partial W$ will each have $6^d$ elements per row. As with SKI, we use FFTs to obtain $\mathcal{O}(m \log m)$ MVMs with $K_{UU}$. Because $W$ and $\partial W$ have $\mathcal{O}(n6^d)$ and $\mathcal{O}(nd6^d)$ nonzero elements, respectively, our MVM complexity is $\mathcal{O}(nd6^d + m \log m)$.

### 3.2 D-SKIP

Several common kernels are *separable*, i.e., they can be expressed as products of one-dimensional kernels. Assuming a compatible approximation scheme, this structure is inherited by the SKI approximation for the kernel matrix without derivatives,

$$K \approx (W_1 K_1 W_1^T) \odot (W_2 K_2 W_2^T) \odot \ldots \odot (W_d K_d W_d^T),$$

where $A \odot B$ denotes the Hadamard product of matrices $A$ and $B$ with the same dimensions, and $W_j$ and $K_j$ denote the SKI interpolation and inducing point grid matrices in the $j$th coordinate direction. The same Hadamard product structure applies to the kernel matrix with derivatives; for example, for $d = 2$,

$$K^\nabla \approx \begin{bmatrix} W_1 K_1 W_1^T & W_1 K_1 \partial W_1^T & W_1 K_1 W_1^T \\ \partial W_1 K_1 W_1^T & \partial W_1 K_1 \partial W_1^T & \partial W_1 K_1 W_1^T \\ W_1 K_1 W_1^T & W_1 K_1 \partial W_1^T & W_1 K_1 W_1^T \end{bmatrix} \odot \begin{bmatrix} W_2 K_2 W_2^T & W_2 K_2 W_2^T & W_2 K_2 \partial W_2^T \\ W_2 K_2 W_2^T & W_2 K_2 W_2^T & W_2 K_2 \partial W_2^T \\ \partial W_2 K_2 W_2^T & \partial W_2 K_2 W_2^T & \partial W_2 K_2 \partial W_2^T \end{bmatrix}. \quad (3)$$

Equation 3 expresses $K^\nabla$ as a Hadamard product of one dimensional kernel matrices. Following this approximation, we apply the SKIP reduction [8] and use Lanczos to further approximate equation 3 as $(Q_1 T_1 Q_1^T) \odot (Q_2 T_2 Q_2^T)$. This can be used for fast MVMs with the kernel matrix. Applied to kernel matrices with derivatives, we call this approach D-SKIP.

Constructing the D-SKIP kernel costs $\mathcal{O}(d^2(n + m \log m + r^3 n \log d))$, and each MVM costs $\mathcal{O}(dr^2 n)$ flops where $r$ is the effective rank of the kernel at each step (rank of the Lanczos decomposition). We achieve high accuracy with $r \ll n$.

### 3.3 Preconditioning

Recent work has explored several preconditioners for exact kernel matrices without derivatives [5]. We have had success with preconditioners of the form $M = \sigma^2 I + F F^T$ where $K^\nabla \approx F F^T$ with $F \in \mathbb{R}^{n \times p}$. Solving with the Sherman-Morrison-Woodbury formula (*a.k.a* the matrix inversion lemma) is inaccurate for small $\sigma$; we use the more stable formula $M^{-1} b = \sigma^{-2}(f - Q_1(Q_1^T b))$ where $Q_1$ is computed in $\mathcal{O}(p^2 n)$ time by the economy QR factorization

$$\begin{bmatrix} F \\ \sigma I \end{bmatrix} = \begin{bmatrix} Q_1 \\ Q_2 \end{bmatrix} R.$$

In our experiments with solvers for D-SKI and D-SKIP, we have found that a truncated pivoted Cholesky factorization, $K^\nabla \approx (\Pi L)(\Pi L)^T$ works well for the low-rank factorization. Computing the pivoted Cholesky factorization is cheaper than MVM-based preconditioners such as Lanczos or truncated eigendecompositions as it only requires the diagonal and the ability to form the rows where pivots are selected. Pivoted Cholesky is a natural choice when inducing point methods are applied as the pivoting can itself be viewed as an inducing point method where the most important information is selected to construct a low-rank preconditioner [12]. The D-SKI diagonal can be formed in $\mathcal{O}(nd6^d)$ flops while rows cost $\mathcal{O}(nd6^d + m)$ flops; for D-SKIP both the diagonal and the rows can be formed in $\mathcal{O}(nd)$ flops.

### 3.4  Dimensionality reduction

In many high-dimensional function approximation problems, only a few directions are relevant. That is, if $f : \mathbb{R}^d \to \mathbb{R}$ is a function to be approximated, there is often a matrix $P$ with $\tilde{d} < d$ orthonormal columns spanning an *active subspace* of $\mathbb{R}^d$ such that $f(x) \approx f(PP^T x)$ for all $x$ in some domain $\Omega$ of interest [4]. The optimal subspace is given by the dominant eigenvectors of the covariance matrix $C = \int_\Omega \nabla f(x) \nabla f(x)^T \, dx$, generally estimated by Monte Carlo integration. Once the subspace is determined, the function can be approximated through a GP on the reduced space, i.e., we replace the original kernel $k(x, x')$ with a new kernel $\check{k}(x, x') = k(P^T x, P^T x')$. Because we assume gradient information, dimensionality reduction based on active subspaces is a natural pre-processing phase before applying D-SKI and D-SKIP.

## 4  Experiments

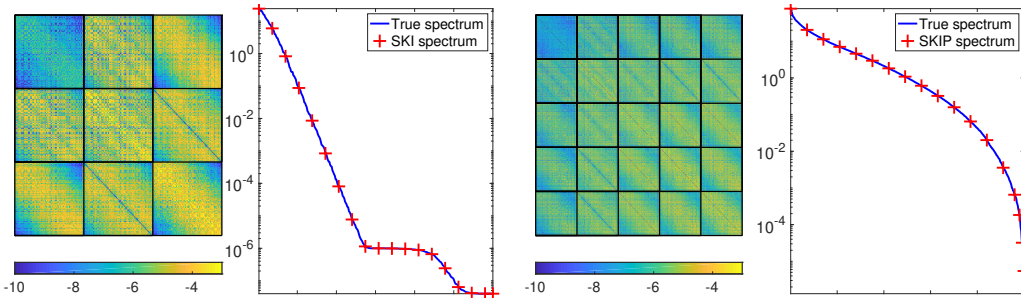

Figure 2:  (Left two images) $\log_{10}$ error in SKI approximation and comparison to the exact spectrum. (Right two images) $\log_{10}$ error in SKIP approximation and comparison to the exact spectrum.

Our experiments use the squared exponential (SE) kernel, which has product structure and can be used with D-SKIP; and the spline kernel, to which D-SKIP does not directly apply. We use these kernels in tandem with D-SKI and D-SKIP to achieve the fast MVMs derived in §3. We write D-SE to denote the exact SE kernel with derivatives.

D-SKI and D-SKIP with the SE kernel approximate the original kernel well, both in terms of MVM accuracy and spectral profile. Comparing D-SKI and D-SKIP to their exact counterparts in Figure 2, we see their matrix entries are very close (leading to MVM accuracy near $10^{-5}$), and their spectral profiles are indistinguishable. The same is true with the spline kernel. Additionally, scaling tests in Figure 3 verify the predicted complexity of D-SKI and D-SKIP. We show the relative fitting accuracy of SE, SKI, D-SE, and D-SKI on some standard test functions in Table 1.

### 4.1  Dimensionality reduction

We apply active subspace pre-processing to the 20 dimensional Welsh test function in [2]. The top six eigenvalues of its gradient covariance matrix are well separated from the rest as seen in Figure 4(a). However, the function is far from smooth when projected onto the leading 1D or 2D active subspace, as Figure 4(b)-4(d) indicates, where the color shows the function value.

We therefore apply D-SKI and D-SKIP on the 3D and 6D active subspace, respectively, using 5000 training points, and compare the prediction error against D-SE with 190 training points because

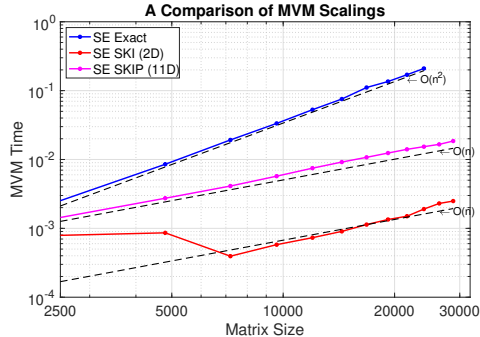

Figure 3: Scaling tests for D-SKI in two dimensions and D-SKIP in 11 dimensions. D-SKIP uses fewer data points for identical matrix sizes.

|       | Branin  | Franke  | Sine Norm | Sixhump | StyTang | Hart3  |
|-------|---------|---------|-----------|---------|---------|--------|
| SE    | 6.02e-3 | 8.73e-3 | 8.64e-3   | 6.44e-3 | 4.49e-3 | 1.30e-2 |
| SKI   | 3.97e-3 | 5.51e-3 | 5.37e-3   | 5.11e-3 | 2.25e-3 | 8.59e-3 |
| D-SE  | 1.83e-3 | 1.59e-3 | 3.33e-3   | 1.05e-3 | 1.00e-3 | 3.17e-3 |
| D-SKI | 1.03e-3 | 4.06e-4 | 1.32e-3   | 5.66e-4 | 5.22e-4 | 1.67e-3 |

Table 1: Relative RMSE error on 10000 testing points for test functions from [24], including five 2D functions (Branin, Franke, Sine Norm, Sixhump, and Styblinski-Tang) and the 3D Hartman function. We train the SE kernel on 4000 points, the D-SE kernel on $4000/(d+1)$ points, and SKI and D-SKI with SE kernel on 10000 points to achieve comparable runtimes between methods.

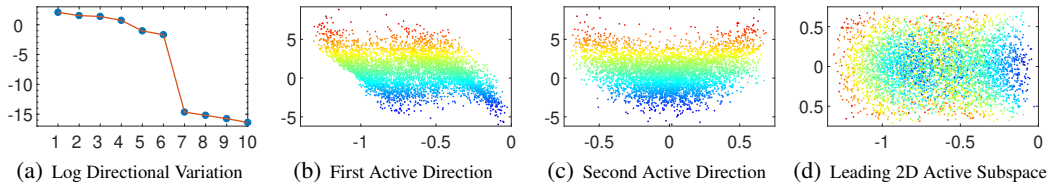

(a) Log Directional Variation    (b) First Active Direction    (c) Second Active Direction    (d) Leading 2D Active Subspace

Figure 4: 4(a) shows the top 10 eigenvalues of the gradient covariance. Welsh is projected onto the first and second active direction in 4(b) and 4(c). After joining them together, we see in 4(d) that points of different color are highly mixed, indicating a very spiky surface.

of our scaling advantage. Table 2 reveals that while the 3D active subspace fails to capture all the variation of the function, the 6D active subspace is able to do so. These properties are demonstrated by the poor prediction of D-SKI in 3D and the excellent prediction of D-SKIP in 6D.

|      | D-SE      | D-SKI (3D) | D-SKIP (6D) |
|------|-----------|------------|-------------|
| RMSE | 4.900e-02 | 2.267e-01  | 3.366e-03   |
| SMAE | 4.624e-02 | 2.073e-01  | 2.590e-03   |

Table 2: Relative RMSE and SMAE prediction error for Welsh. The D-SE kernel is trained on $4000/(d+1)$ points, with D-SKI and D-SKIP trained on 5000 points. The 6D active subspace is sufficient to capture the variation of the test function.

## 4.2 Rough terrain reconstruction

Rough terrain reconstruction is a key application in robotics [9, 15], autonomous navigation [10], and geostatistics. Through a set of terrain measurements, the problem is to predict the underlying topography of some region. In the following experiment, we consider roughly 23 million non-uniformly sampled elevation measurements of Mount St. Helens obtained via LiDAR [3]. We bin the measurements into a $970 \times 950$ grid, and downsample to a $120 \times 117$ grid. Derivatives are approximated using a finite difference scheme.

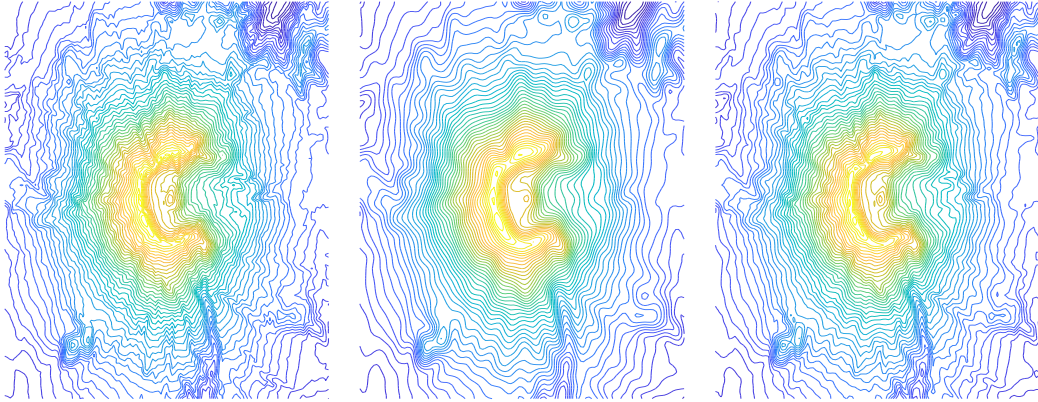

Figure 5: On the left is the true elevation map of Mount St. Helens. In the middle is the elevation map calculated with the SKI. On the right is the elevation map calculated with D-SKI.

We randomly select $90\%$ of the grid for training and the remainder for testing. We do not include results for D-SE, as its kernel matrix has dimension roughly $4 \cdot 10^4$. We plot contour maps predicted by SKI and D-SKI in Figure 5 — the latter looks far closer to the ground truth than the former. This is quantified in the following table:

|  | $\ell$ | $s$ | $\sigma$ | $\sigma_2$ | Testing SMAE | Overall SMAE | Time[s] |
|---|---|---|---|---|---|---|---|
| SKI | 35.196 | 207.689 | 12.865 | n.a. | 0.0308 | 0.0357 | 37.67 |
| D-SKI | 12.630 | 317.825 | 6.446 | 2.799 | 0.0165 | 0.0254 | 131.70 |

Table 3: The hyperparameters of SKI and D-SKI are listed. Note that there are two different noise parameters $\sigma_1$ and $\sigma_2$ in D-SKI, for the value and gradient respectively.

## 4.3 Implicit surface reconstruction

Reconstructing surfaces from point cloud data and surface normals is a standard problem in computer vision and graphics. One popular approach is to fit an implicit function that is zero on the surface with gradients equal to the surface normal. Local Hermite RBF interpolation has been considered in prior work [17], but this approach is sensitive to noise. In our experiments, using a GP instead of splining reproduces implicit surfaces with very high accuracy. In this case, a GP with derivative information is required, as the function values are all zero.

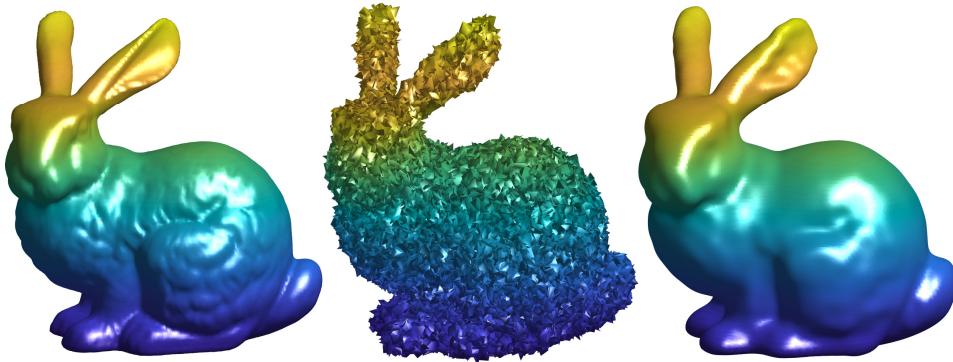

Figure 6: (Left) Original surface (Middle) Noisy surface (Right) SKI reconstruction from noisy surface ($s = 0.4, \sigma = 0.12$)

In Figure 6, we fit the Stanford bunny using 25000 points and associated normals, leading to a $K^\nabla$ matrix of dimension $10^5$, clearly far too large for exact training. We therefore use SKI with the thin-plate spline kernel, with a total of 30 grid points in each dimension. The left image is a ground truth mesh of the underlying point cloud and normals. The middle image shows the same mesh, but with heavily noised points and normals. Using this noisy data, we fit a GP and reconstruct a surface shown in the right image, which looks very close to the original.

### 4.4 Bayesian optimization with derivatives

Prior work examines Bayesian optimization (BO) with derivative information in low-dimensional spaces to optimize model hyperparameters [29]. Wang et al. consider high-dimensional BO (without gradients) with random projections uncovering low-dimensional structure [26]. We propose BO with derivatives and dimensionality reduction via active subspaces, detailed in Algorithm 1.

---

**Algorithm 1: BO with derivatives and active subspace learning**

---

1: **while** Budget not exhausted **do**
2:     Calculate active subspace projection $P \in \mathbb{R}^{d \times \tilde{d}}$ using sampled gradients
3:     Optimize acquisition function, $u_{n+1} = \arg\max \mathcal{A}(u)$ with $x_{n+1} = P u_{n+1}$
4:     Sample point $x_{n+1}$, value $f_{n+1}$, and gradient $\nabla f_{n+1}$
5:     Update data $\mathcal{D}_{i+1} = \mathcal{D}_i \cup \{x_{n+1}, f_{n+1}, \nabla f_{n+1}\}$
6:     Update hyperparameters of GP with gradient defined by kernel $k(P^T x, P^T x')$
7: **end**

---

Algorithm 1 estimates the active subspace and fits a GP with derivatives in the reduced space. Kernel learning, fitting, and optimization of the acquisition function all occur in this low-dimensional subspace. In our tests, we use the expected improvement (EI) acquisition function, which involves both the mean and predictive variance. We consider two approaches to rapidly evaluate the predictive variance $v(x) = k(x,x) - K_{xX}\tilde{K}^{-1}K_{Xx}$ at a test point $x$. In the first approach, which provides a biased estimate of the predictive variance, we replace $\tilde{K}^{-1}$ with the preconditioner solve computed by pivoted Cholesky; using the stable QR-based evaluation algorithm, we have

$$v(x) \approx \hat{v}(x) \equiv k(x,x) - \sigma^{-2}(\|K_{Xx}\|^2 - \|Q_1^T K_{Xx}\|^2).$$

We note that the approximation $\hat{v}(x)$ is always a (small) overestimate of the true predictive variance $v(x)$. In the second approach, we use a randomized estimator as in [1] to compute the predictive variance at many points $X'$ simultaneously, and use the pivoted Cholesky approximation as a control variate to reduce the estimator variance:

$$v_{X'} = \text{diag}(K_{X'X'}) - \mathbb{E}_z \left[ z \odot (K_{X'X}\tilde{K}^{-1}K_{XX'}z - K_{X'X}M^{-1}K_{XX'}z) \right] - \hat{v}_{X'}.$$

The latter approach is unbiased, but gives very noisy estimates unless many probe vectors $z$ are used. Both the pivoted Cholesky approximation to the predictive variance and the randomized estimator resulted in similar optimizer performance in our experiments.

To test Algorithm 1, we mimic the experimental set up in [26]: we minimize the 5D Ackley and 5D Rastrigin test functions [24], randomly embedded respectively in $[-10, 15]^{50}$ and $[-4, 5]^{50}$. We fix $\tilde{d} = 2$, and at each iteration pick two directions in the estimated active subspace at random to be our active subspace projection $P$. We use D-SKI as the kernel and EI as the acquisition function. The results of these experiments are shown in Figure 7(a) and Figure 7(b), in which we compare Algorithm 1 to three other baseline methods: BO with EI and no gradients in the original space; multi-start BFGS with full gradients; and random search. In both experiments, the BO variants perform better than the alternatives, and our method outperforms standard BO.

## 5   Discussion

When gradients are available, they are a valuable source of information for Gaussian process regression; but inclusion of $d$ extra pieces of information per point naturally leads to new scaling issues. We introduce two methods to deal with these scaling issues: D-SKI and D-SKIP. Both are structured

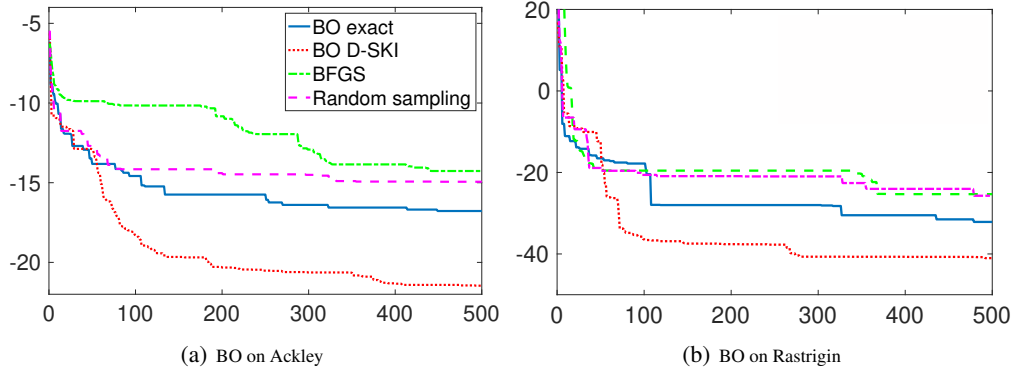

<div align="center">(a) BO on Ackley        (b) BO on Rastrigin</div>

Figure 7: In the following experiments, 5D Ackley and 5D Rastrigin are embedded into 50 a dimensional space. We run Algorithm 1, comparing it with BO exact, multi-start BFGS, and random sampling. D-SKI with active subspace learning clearly outperforms the other methods.

interpolation methods, and the latter also uses kernel product structure. We have also discussed practical details —preconditioning is necessary to guarantee convergence of iterative methods and active subspace calculation reveals low-dimensional structure when gradients are available. We present several experiments with kernel learning, dimensionality reduction, terrain reconstruction, implicit surface fitting, and scalable Bayesian optimization with gradients. For simplicity, these examples all possessed full gradient information; however, our methods trivially extend if only partial gradient information is available.

There are several possible avenues for future work. D-SKIP shows promising scalability, but it also has large overheads, and is expensive for Bayesian optimization as it must be recomputed from scratch with each new data point. We believe kernel function approximation via Chebyshev interpolation and tensor approximation will likely provide similar accuracy with greater efficiency. Extracting low-dimensional structure is highly effective in our experiments and deserves an independent, more thorough treatment. Finally, our work in scalable Bayesian optimization with gradients represents a step towards the unified view of global optimization methods (i.e. Bayesian optimization) and gradient-based local optimization methods (i.e. BFGS).

**Acknowledgements.** We thank NSF DMS-1620038, NSF IIS-1563887, and Facebook Research for support.

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
