[Supplementary Material]

# Supplementary material: Scaling Gaussian Process Regression with Derivatives

**David Eriksson**
Center for Applied Mathematics
Cornell University
Ithaca, NY 14853
dme65@cornell.edu

**Kun Dong**
Center for Applied Mathematics
Cornell University
Ithaca, NY 14853
kd383@cornell.edu

**Eric Hans Lee**
Department of Computer Science
Cornell University
Ithaca, NY 14853
ehl59@cornell.edu

**David Bindel**
Department of Computer Science
Cornell University
Ithaca, NY 14853
bindel@cornell.edu

**Andrew Gordon Wilson**
Department of Operations Research
and Information Engineering
Cornell University
Ithaca, NY 14853
agw62@cornell.edu

## 1 Kernels

The covariance functions we consider in this paper are the squared exponential (SE) kernel

$$k_{\text{SE}}(x, y) = s^2 \exp\left(-\frac{\|x - y\|^2}{2\ell^2}\right)$$

and the spline kernels

$$k_{\text{spline}}(x, y) = \begin{cases} s^2\left(\|x - y\|^3 + a\|x - y\|^2 + b\right) & d \text{ odd} \\ s^2\left(\|x - y\|^2 \log\|x - y\| + a\|x - y\|^2 + b\right) & d \text{ even} \end{cases}$$

where $a, b$ are chosen to make the spline kernel symmetric and positive definite on the given domain.

## 2 Kernel Derivatives

The first and second order derivatives of the SE kernel are

$$\frac{\partial k_{\text{SE}}(x^{(i)}, x^{(j)})}{\partial x_p^{(j)}} = \frac{x_p^{(i)} - x_p^{(j)}}{\ell^2} k_{\text{SE}}(x^{(i)}, x^{(j)}),$$

$$\frac{\partial^2 k_{\text{SE}}(x^{(i)}, x^{(j)})}{\partial x_p^{(i)} \partial x_q^{(j)}} = \frac{1}{\ell^4}\left(\ell^2 \delta_{pq} - (x_p^{(i)} - x_p^{(j)})(x_q^{(i)} - x_q^{(j)})\right) k_{\text{SE}}(x^{(i)}, x^{(j)}).$$

This shows that each $n$-by-$n$ block of $\partial K$ and $\partial^2$ admit Kronecker and Toeplitz structure if the points are on a grid.

# 3 Preconditioning

We discover that preconditioning is crucial for the convergence of iterative solvers using approximation schemes such as D-SKI and D-SKIP. To illustrate the performance of conjugate gradient (CG) method with and without the above-mentioned truncated pivoted Cholesky preconditioner, we test D-SKI on the 2D Franke function with 2000 data points, and D-SKIP on the 5D Friedman function with 1000 data points. In both cases, we compute a pivoted Cholesky decomposition truncated at rank 100 for preconditioning, and the number of steps it takes for CG/PCG to converge are demonstrated in Figure 1 below. It is clear that preconditioning universally and significantly reduces the number of steps required for convergence.

Figure 1: The color shows $\log_{10}$ of the number of iterations to reach a tolerance of 1e-4. The first row compares D-SKI with and without a preconditioner. The second row compares D-SKIP with and without a preconditioner. The red dots represent no convergence. The y-axis shows $\log_{10}(\ell)$ and the x-axis $\log_{10}(\sigma)$ and we used a fixed value of $s = 1$.

# 4 Korea

(a) Ground Truth      (b) SKI      (c) D-SKI

Figure 2: D-SKI is clearly able to capture more detail in the map than SKI. Note that inclusion of derivative information in this case leads to a negligible increase in calculation time.

The Korean Peninsula elevation and bathymetry dataset[1] is sampled at a resolution of 12 cells per degree and has $180 \times 240$ entries on a rectangular grid. We take a smaller subgrid of $17 \times 23$ points as training data. To reduce data noise, we apply a Gaussian filter with $\sigma_{\text{filter}} = 2$ as a pre-processing step. We observe that the recovered surfaces with SKI and D-SKI highly resemble their respective counterparts with exact computation and that incorporating gradient information enables us to recover more terrain detail.

|  | $\ell$ | $s$ | $\sigma$ | SMAE | Time[s] |
|---|---|---|---|---|---|
| SKI | 16.786 | 855.406 | 184.253 | 0.1521 | 10.094 |
| D-SKI | 9.181 | 719.376 | 29.486 | 0.0746 | 11.643 |

# References

[1] MATLAB mapping toolbox, 2017. The MathWorks, Natick, MA, USA.