[Reviews · NeurIPS 2018]

Reviewer 1



The paper seems well-written and technically sound. I suggest to improve the presentation and the clarity of section 3.1.

Reviewer 2



The paper investigates the application of the recent SKI and SKIP GP methodologies in the case where additional gradient information is available. They derive the necessary mathematical background, discuss numerical strategies to make the computations robust efficient and efficient and evaluate extensively on established synthetic and real benchmark problems. A weakness of the paper is the somewhat entangled discussion of dimensionality reduction strategies. D-SKI scales with 6^d, which is approximately 60 million for d=10 and roughly half a trillion for d=15. These numbers demonstrate that D-SKIP will not be optional for many problems typically considered with BO. From the presented experiments, it is unclear to me how the "online dimension reduction" would influence the BO algorithm. Additionally, I find the description in lines 205-211 too vague to be able to understand the experimental evaluation. I will not consider this a problem for the submission as long as the authors hold up their promise of providing the experiment code. (As a minor side note: you might want to remind the reader of reference [25] at this point in the text.) Nevertheless, a strength of the paper is its holistic description and evaluation of the proposed method. The background and the newly extended analysis seem straight forward, so the authors make good use of the page limit to discuss important practical strategies like preconditioning and provide a principled evaluation. Their basic experiments are run on common test functions which could in principle be used for any desired mathematical analysis. I agree with their rationale on terrain and surface reconstruction and the experiments are convincing. In terms of significance, I am hesitant to provide a clear judgement. Personally, I have not worked on surface reconstruction before, so I don't know how big this application area is. In terms of originality, this seems to be a clear continuation from early work on SKI and SKIP. As a consequence, I am not surprised by this work, but I can imagine that providing this detailed derivation (and, possibly, the code) was still a highly non-trivial issue, in particular with respect to the pre-conditioning evaluation and experiments. In conclusion: in my opinion, this is a solid piece of research, whose lack of ground-braking novelty is made up by the good presentation and experimental evaluation. ---- Post-rebuttal update: Thanks for the authors for the clarifications. I very much appreciated your comment regarding the computational complexity of D-SKIP and I encourage the authors to put this information into the paper as well. I am willing to follow your arguments regarding significance and novelty, and I will support this view in the discussion. I have not updated my score of the overall review as this score already reflected my ignorance in terms of significance "in dubio pro reo".

Reviewer 3



The authors present a method of scaling GP regression that takes advantage of the derivative information of the function. Scalability in the number of samples and dimensions is achieved by approximation of the kernel matrix using structured kernel interpolation (SKI) and SKI for products (SKIP) methods. The authors demonstrate the promise of their method on several numerical examples. The method is extended for Bayesian optimization by using dimension reduction via subspace projection. The method is simple and combines several well-known methods for better and scalable GP regression. The manuscript is well-written and numerical examples are diverse. Major issues: - A major criticism of this paper is that the work in incremental. The authors rely on SKI and SKIP for scalability. The use of derivatives (first and second) is well known in Bayesian uncertainty quantification literature and is a special case of multi-output GP. Finally, the dimension reduction approach is also well-known. This work is good but the authors fail to emphasize the "NIPS-level" novelty of this work. - No guarantees are provided in terms of the quality of approximation. In particular, how does the approximation error decay as a function of n, d, number of inducing points, and smoothness of the function. - How do the authors propose to use their method where the function is non-differentiable? Some of the most widely used GPs are non-differentiable; for example, Brownian motion.